# Studies on Annihilation and Coreactant Electrochemiluminescence of Thermally Activated Delayed Fluorescent Molecules in Organic Medium

**DOI:** 10.3390/molecules27217457

**Published:** 2022-11-02

**Authors:** Ping Huang, Xingzi Zou, Zhiyun Xu, Yanting Lan, Lijuan Chen, Baohua Zhang, Li Niu

**Affiliations:** Center for Advanced Analytical Science, Guangzhou Key Laboratory of Sensing Materials & Devices, School of Chemistry and Chemical Engineering, Guangzhou University, Guangzhou 510006, China

**Keywords:** electrochemiluminescence, thermally activated delayed fluorescence (TADF), coreactant ECL, ECL sensing, spectra

## Abstract

Very recently, there is a great research interest in electrochemiluminescence (ECL) featuring thermally activated delayed fluorescence (TADF) properties, i.e., TADF-ECL. It is appealing since the earlier reports in this topic well-confirmed that this strategy has a great potential in achieving all-exciton-harvesting ECL efficiency under electrochemical excitation, which is a breakthrough in the topic of organic ECL. However, organic phase electrochemistry and ECL studies surrounding TADF-ECL are still extremely rare. Especially, the ECL spectra of previous reported TADF emitters are still very different from their PL spectra. In this work, we systematically measure and discuss the liquid electrochemistry and ECL behavior of two typical TADF molecules in organic medium. Most importantly, we verify for the first time that the ECL spectra of them (coreactant ECL mode) are identical to their PL spectra counterparts, which confirms the effectiveness of TADF photophysical properties in the coreactant ECL mode in practice.

## 1. Introduction

It is well-known that electrochemiluminescence (ECL) is one promising analytical method, in which photon signals are electrogenerated by electrochemical excitation in electrolytic cells [1,2]. Up to the present, a great deal of applications has been realized for such ECL techniques, such as ultrasensitive life analysis, environmental analysis, high-resolution activity mapping on nanocatalysts and more recently single-photon-level tissue and cell imaging, etc. [1,2,3,4,5,6,7]. Despite these successes, it is obvious in that the scarcity of qualified ECL luminophores largely hampers the development of ECL. Until now, except for some peculiar cases [8], almost all ECL applications use the state-of-the-art tris(2,2′-bipyridyl)ruthenium(II) Ru(bpy)_3_^2+^ or its analogues [2,7,9]. This system has been well-applied since it possesses satisfactory electrochemical stability, high ECL efficiency and mature labelling methodology towards various analyzing targets [9]. However, there are still so many drawbacks to be resolved for this system, such as the high potential for ECL, hard to tailor ECL spectra, high cost and very limited room for further enhancing of ECL efficiency (Φ_ECL_). To accelerate the development of ECL, it is excepted to enrich qualified ECL luminophores. In this way, much better overall ECL performance is anticipated.

Nowadays, alongside such demands, a great deal of attention is focusing on developing advanced organic ECL luminophores [10,11,12,13,14,15]. Especially, two peculiar strategies have been launched with great developing potential [16], that is: (i) organic aggregation-induced ECL (AIECL) [17] and (ii) thermally activated delayed fluorescent ECL (TADF-ECL) [18,19], respectively. As for AIECL [20], it is appealing since the notorious aggregation-caused quenching (ACQ) limiting issue is largely restrained or even removed by the aggregation-induced emission (AIE) effect [21], which guarantees the achievements of a much higher Φ_ECL_ for them [10,12,17,22,23,24]. For instance, a series of organic AIECL systems were developed, e.g., tetraphenylethylene (TPE)-based nanocrystals or polymer dots [17,24], carbon dots [23], silole or triphenylporphyrin-based compounds [10,22]. Most of them showed excellent ECL stabilities and satisfactory Φ_ECL_ and even were applied in various biosensing applications with satisfactory effects [14,15]. As for the topic of TADF-ECL [18,19], it is significant in that such a strategy is characteristic of all-exciton-harvesting for ECL emission. Compared to every reported organic ECL system [17,22,25,26,27], the theoretical Φ_ECL_ of organic TADF-ECL systems is increased by a factor of four, i.e., from 25% to 100%. It is reasonable since common organic ECL systems belong to fluorescence in photophysics. In this case, it is merely electrochemically generated singlets (ca. 25% in total) that are harvested for ECL emissions. At the best condition, i.e., when the PL quantum efficiency (Φ_PL_) of those ECL fluorophores equals to 100%, the maximized Φ_ECL_ of those fluorescent-type organic ECL systems can achieve a value of 25%. By contrast, TADF-ECL outperforms in Φ_ECL_ since all those dark triplets (~75% in total under electrochemical excitation) [18] can be emissive through the delayed fluorescent route, i.e., DF-ECL (Figure 1). Such processes can be efficient since the exchange energy (ΔE_ST_) between the first singlet level (S_1_) and the first triplet level (T_1_) is low enough (0.1–0.3 eV) for TADF luminophores [28], which guarantees the achievement of a fast and efficient reverse inter-system crossing (RISC) process from T_1_ to S_1_ (Figure 1). In this case, the maximized Φ_ECL_ of the TADF-ECL system can be as high as 100%.

To date, some progress has been achieved in the topic of TADF-ECL, such as the achievements of highly efficient TADF-ECL in organic medium [18,19,29,30,31], extremely narrow ECL spectra [32], reliable TADF-ECL in aqueous medium [12,33] and TADF-ECL sensing applications [19,34]. In 2014, Ishimatsu et al. reported the first annihilation TADF-ECL, in which a TADF molecular 1,2,3,5-tetrakis(carbazol-9-yl)-4,6-dicyanobenzene (4CzIPN) [35] was used as the ECL luminophore in organic medium [18]. Under high-frequency step potential driving, it achieved a Φ_ECL_ of 47 ± 6.0% in dichloromethane (DCM) medium, which approached the corresponding Φ_PL_, i.e., 54%. The ECL spectra of the annihilation ECL system containing 4CzIPN resembled the PL spectra of 4CzIPN. These results were combined to confirm the achievement of TADF-ECL in annihilation ECL mode [18,29]. To achieve the practical sensing application of TADF-ECL in real scenarios, our group restarted the studies of TADF-ECL in 2021 [12,19,30,33] and focused on the possibility of coreactant TADF-ECL. As confirmed, stable and efficient oxidative–reduction polymer TADF-ECL [19] or reductive–oxidation polymer TADF-ECL [30] was successfully constructed by using TADF polymer modified glassy carbon electrode (GCE) as the light-emissive surface-modified working electrode of the coreactant TADF-ECL system. Either of them showed much higher Φ_ECL_ (nearly four-fold) as compared to that of the traditional fluorescent ECL counterparts. Under successive electrochemical driving (step potential or linear CV scanning), the ECL stability of those TADF-ECL systems is also satisfactory. Solid-state TADF-ECL sensing on L-cysteine was further performed, showing ultralow detection limits, high sensitivity and good specificity [19]. Very recently, we also realized stable and efficient aqueous TADF-ECL [12,33,34], which is significant to achieve practical TADF-ECL sensing applications in life science in the near future. The corresponding methods are divided into two general categories. Firstly, we have developed TADF-ECL in aqueous media by using aggregation-induced delayed fluorescence (AIDF) luminogens, also called as AIDF-ECL [12]. Since AIDF-ECL integrates the merits of TADF and the AIE effect of organic luminophores in aqueous media, the Φ_ECL_ of such an AIDF-ECL model system distinctly outperformed that of a TPE-based AIECL counterpart. Secondly, we have explored the nanoencapsulation strategy to prepare air-stable and water-soluble TADF-ECL luminophores [33]. Importantly, the oxygen quenching effect [36] on the TADF-ECL system in an aqueous medium is well-removed by this method. Using those aqueous-soluble TADF molecular nanoparticles such as ECL luminophores, stable and efficient aqueous TADF-ECL were realized, irrespective of annihilation or coreactant ECL [33]. In early 2022, using such nanoencapsulation strategies, we reported on the first aqueous coreactant TADF-ECL dopamine biosensing applications, which showed satisfactory linearity, selectivity, repeatability and detection limits [34].

Despite those progresses in the topic of TADF-ECL, it should be noticed that understanding of mechanisms of TADF-ECL is still in its infancy. Especially, as for the coreactant ECL mode, we found that the ECL spectra of the reported TADF luminophores/coreactant couple do not fully resemble its PL spectra counterparts [12,19,30,31,33,34]. It is plausible in that in some cases, more complicated mechanisms could be involved, e.g., the exciplex formation and organic long-persistent emission [31]. To confirm the coreactant TADF-ECL definitively, it is believed that the corresponding ECL and PL spectra of a certain TADF luminophores/coreactant couple should be the same. Moreover, the questions concerning the relationship between the electrochemical properties of TADF luminophores and its ECL efficiency, stability and potential should be further studied. To answer these questions, it is necessary to further perform electrochemical and ECL studies of typical TADF molecules in an organic medium, which is the most simplified condition to disclose such questions.

Herein, we present the detailed studies on the annihilation and coreactant ECL of two typical TADF molecules in a DCM medium. First of all, basic photophysical and electrochemical measurements are performed to clarify their intrinsic physical and electrochemical properties. After that, annihilation and coreactant ECL studies are conducted, including the evaluation of Φ_ECL_, ECL stability, potentials and most importantly their ECL spectra. Very meaningfully, the ECL and PL spectra of those two TADF luminophores in the coreactant ECL mode are identical, which confirms the TADF emission nature of those luminophores in the coreactant ECL mode. Moreover, some peculiar clues were discovered to understand the determining factors of Φ_ECL_ and ECL potentials of coreactant TADF-ECL, which is very meaningful to enrich our understanding on TADF-ECL and accelerate its development towards higher performance and better applications.

## 2. Experimental

### 2.1. TADF Luminophores

The TADF compound of 4CzIPN (788.89 g mol^−1^) [35] was purchased from Xi’an Polymer Light Technology Corp, while the homemade 3,6-di(tert-butyl)-1,8-di(4-(bis(4-(tertbutyl)phenyl)amino)phenyl)-9-(4-(4,6-diphenyl-1,3,5-triazin-2-yl) phenyl) carbazole (BPAPTC) (1296.77 g mol^−1^) [37] was synthesized by our collaborators. All those materials were used as received.

### 2.2. Measurements of Photophysics

All those photophysical studies were performed on DCM solution containing those TADF molecules with a concentration of 25 μM. The absorbance was measured by a UV–Vis spectrophotometer (Shimadzu UV-1780, Shimadzu, Kyoto, Japan). The steady-state PL spectra and transient PL decay curves of those samples in solution were conducted by the Edinburgh FLS1000 spectrofluorometer, in which an Xe2 xenon lamp and a picosecond pulsed LED (EPLED-365) were the light source (365 nm as the excitation wavelength) while it was required. The PL transient delay curves were fitted by bi-exponential functions, which was commonly used to derive the prompt and delayed fluorescent emissions of TADF emitters [38]. The absolute PLQY in the atmosphere of those samples was measured by an integrating sphere that was coupled with Edinburgh FLS1000.

### 2.3. Cyclic Voltammetry (CV) and ECL Measurements

CV experiments were measured by the CHI 660B electrochemistry workstation (CH Instruments Inc.). Prior to CV measurements, the glassy carbon electrode (GCE) working electrode (4 mm in diameter) was routinely cleaned [12]. A common three-electrode configuration was used for those CV studies, in which GCE, Pt wire and Ag wire were the working electrode, counter electrode and the quasi-reference electrode, respectively (0.1 mM 4CzIPN or 0.1 mM BPAPTC molecules dissolved in 0.1 M tetra-*n*-butylammonium hexafluorophosphate (TBAPF_6_) as supporting electrolyte in DCM). The CV tests used the scanning rate of 100 mV/s and ferrocene (Fc)/ferrocenium (Fc^+^) as the calibrating reference [39].

ECL studies were measured by the MPI–EII ECL detection system (Remex Electronic Instrument Lt. Co, Xi’an, China), in which the configurations and the condition of the electrolytic cells are the same as that measured in CV studies. The more detailed introductions were presented in our earlier work [19]. The PMT voltage and scanning rate were set at 850 V and 100 mV/s, respectively. For the oxidative–reduction ECL and reductive–oxidation ECL studies in DCM, the used TADF luminophores were 0.1 mM in concentration and the coreactant was tri-n-propylamine (TPrA) (40 mM) or benzoyl peroxide (BPO) (25 mM), respectively. While for determining the relative Φ_ECL_, these coupled coreactant ECL systems were measured by using the general method [19,40], i.e., using the sample of Ru(bpy)_3_^2+^ (0.1 mM)/TPrA (40 mM) in acetonitrile solution containing 0.1 M TBAPF_6_ as the reference. The ECL spectra of these were measured by Edinburgh FLS1000, in which the electrolytic cells were placed into the sample chamber of Edinburgh FLS1000 and electrochemically triggered by the CHI 660B electrochemistry workstation.

## 3. Results and Discussion

### 3.1. Photophysical Properties of TADF Luminophores in DCM

Previously, 4CzIPN [35] and BPAPTC [37] were confirmed to be satisfactory TADF emitters and well-applied in organic light-emitting diodes. As a typical charge-transfer-type emitter [41], photophysical properties of these TADF molecules are largely influenced by the solvent that it is dissolved in. Since the subsequent CV and ECL studies are performed in the DCM solvent, we measured the corresponding photophysical properties in DCM. As it is shown in Figure 1a,c, the absorption spectra of those two emitters are basically unchanged as compared to the earlier reports measured in toluene [42]. Either of them show their intrinsic absorption and clear charge-transfer absorption features at the long wavelength range, indicating that the charge-transfer features of these two molecules are maintained in DCM. As for the steady-state PL spectra, some extent of redshift and broadening are observed, e.g., the PL emission peak (λ_PL_) of 4CzIPN and BPAPTC in DCM locates at 543 and 585 nm, respectively, rather than 507 and 520 nm in toluene [35,37]. Such a difference is commonly reported for TADF materials and ascribed to the polarity effect of the solvent medium [41]. In this way, the PL spectra of TADF luminophores are gradually redshifted on increasing polarity of solvents. Figure 1b,d showed their respective PL transient behaviors, which displayed the typical two-component PL transient behaviors of TADF emitters [35]. According to the well-known photophysical theory of TADF emission [28,38,43], the corresponding prompt and delayed fluorescent lifetime (τ_pf_, τ_df_) and ratios (Φ_pf_/Φ_df_) were calculated, i.e., 24 ns, 1603 ns, 39.1%/60.9% for 4CzIPN and 30 ns, 328 ns, 63.5%/36.5% for BPAPTC, respectively, surely confirming the TADF emission features of those two samples in DCM. The measured conditions of those PL transient experiments are the same as that used in the subsequent CV/ECL study. Under electrochemical driving in those DCM media, it is anticipated that these two luminophores will emit light via the same TADF mechanism.

### 3.2. Electrochemistry

CV was performed for those two molecules in DCM solvent to determine their redox properties, in which a routine three-electrode structure was used, i.e., a GCE as the working electrode, a Pt wire as the counter electrode and an Ag wire as the quasi-reference electrode in DCM media containing 0.1 M TBAPF_6_ as a supporting electrolyte with a scanning rate of 100 mV/s. As shown in Appendix A, 4CzIPN displayed a clear reversible cathodic wave with an onset reduction potential at −1.29 V (vs. Ag/Ag+) but irreversible anodic wave with an onset oxidation potential at +1.28 V (vs. Ag/Ag+). It is analogous to the earlier results measured in acetonitrile solvent [33]. Accordingly, it indicates that the electroreduction process of 4CzIPN should be reversible and stable while the electrooxidation process is unstable. As mentioned [18,27], the electrochemical oxidation of carbazole would be involved for such an electrooxidation process. As for the CV results of BPAPTC molecules in DCM (shown in Appendix A), it is different in that in this situation, the electrooxidation of BPAPTC is becoming reversible and shows a clear reversible wave with an onset oxidation potential at 0.66 V (vs. Ag/Ag+). There is no distinctive electrochemical reduction wave for BPAPTC, indicating that the corresponding electroreduction process is unstable.

### 3.3. Annihilation and Coreactant TADF-ECL

At first, annihilation ECL studies of these two TADF luminophores were performed. As shown in Figure 2a, under linear CV scanning, intense ECL emission can be observed for 4CzIPN either in the anodic or cathodic bias condition, which indicates that either radical electrooxidation or electroreduction products of 4CzIPN, i.e., 4CzIPN^●+^ or 4CzIPN^●−^, are stable to guarantee exciton formation and then followed by ECL emission. Meanwhile, the ECL onset potentials, irrespective of anodic or cathodic scanning ranges, resemble its redox potentials showing in Appendix A. We also notice that the anodic ECL intensity of 4CzIPN is higher than its cathodic ECL intensity. This result is consistent with its CV behaviors. Accordingly, it is highly possible that the stability of 4CzIPN^●−^ is superior to that of 4CzIPN^●+^. Under step potential operation (±1.6V, 1Hz), the ECL intensity of them (either at +1.6 V or −1.6 V) is distinctly enhanced by ca. 3-fold and becomes comparable. As compared to CV linear scanning (Figure 2a), the step potential driving condition (Figure 2b) shortens the waiting time of those radical intermediate products prior to collision, which accounts for such enhancing effects of ECL intensity. As for BPAPTC, it is different in that the cathodic ECL intensity is dramatically higher than the anodic ECL intensity (Figure 2c). Compared to 4CzIPN (Figure 2a), the anodic and cathodic ECL onset potentials of BPAPTC (Figure 2c) were changed to ca. 0.6 V and −1.8 V, respectively, which is also consistent with its electrochemical redox characteristics shown in Appendix A. Under step potential driving between +0.8 V and −2 V (1 Hz), the cathodic ECL intensity at −2 V is also dramatically enhanced while the anodic ECL intensity at +0.8 V is still weak (Figure 2d). Accordingly, it indicates that under the electroreduction process, the radical intermediate species of BPAPTC, i.e., BAPTC^●−^, are highly unstable, while the BAPTC^●+^ counterpart generated under the electrooxidation process is stable enough. Such deduction is also consistent with the CV characteristics (Appendix A). In short, in conjunction with CV studies, the corresponding annihilation ECL studies of these two TADF molecules well-reflect the intrinsic relationship between its CV characteristics and ECL potential, intensity and stability. To achieve highly intensive ECL emissions in annihilation ECL mode, these TADF molecules should possess satisfactory electrochemical reversibility.

Subsequently, coreactant ECL studies are conducted for these two TADF molecules, in which TADF molecular 4CzIPN or BPAPTC in couple with the state-of-the-art coreactant, i.e., TPrA or BPO, are dissolved in DCM medium containing 0.1 M TBAPF_6_ supporting electrolyte and a scanning rate of 100 mV/s is used to perform the corresponding ECL measurements (see Experimental Sections for the details). First of all, the results of oxidative–reduction ECL using 0.1 mM 4CzIPN/40 mM TPrA are shown in Figure 3. As depicted, compared to the condition using the bare GCE as the working electrode, the addition of 40 mM TPrA in the electrolytic cells sharply enhances the electrochemical oxidation current. Especially, while the potential is higher than +0.61 V (vs. Ag/Ag+), the anodic current is obviously increased. According to earlier reports [44,45], it corresponds to the electrooxidation process of TPrA. During this scanning ranging from 0 to 1.6 to 0 V, no detectable ECL signals can be observed (Figure 3b). Additionally, the electrooxidation of 0.1 mM 4CzIPN alone does not lead to ECL emission (not shown here). By contrast, as for the situation involving the couple of 0.1 mM 4CzIPN/40 mM TPrA, the anodic current is distinctly increased while the potential is higher than +0.64 V and reaches the peak current at ca. +1.19 V (vs. Ag/Ag+). Therefore, in this case, both TPrA and 4CzIPN are electrochemically oxidized. Meanwhile, the ECL intensity is gradually enhanced as the potential is higher than ca. 1.2 V and reaches the maximum value at +1.6 V. These results are combined to confirm the occurrence of coreactant ECL, in which the oxidized TPrA functions as the reducing agent to inject electrons into the oxidized 4CzIPN radical intermediate species, i.e., 4CzIPN^●+^, to generate excitons, which is followed by ECL emission via radiative decay. We further characterize the CV/ECL stability via 36 multicycle anodic scanning of this system containing the 4CzIPN/TPrA couple. As it is shown in Figure 3c, the anodic current is stable under these continuous cycle scanning conditions. Meanwhile, after the first seven cycles, the ECL intensity is also somewhat stable (Figure 3d). We notice that the ECL stability of such oxidative–reduction ECL mode of 4CzIPN is distinctly better than that observed in annihilation ECL mode [18,29]. In that report, the inherently unstable 4CzIPN^●+^ largely limited the corresponding ECL stability. Our results indicate that such an unstable issue can be solved to a large extent by adding an effective coreactant, e.g., TPrA. In this way, fast and efficient electron transfer processes between the coreactant, e.g., TPrA, and the unstable radical intermediate species of TADF luminophores, e.g., 4CzIPN^●+^, could accelerate the process of exciton formation on TADF luminophores, which is critical to enhance its ECL intensity and stability for practical sensing applications.

Reductive–oxidation ECL studies on 4CzIPN are further conducted by using the traditional BPO as the coreactant. As shown in Figure 4a, no ECL signals can be detected while the electrolytic cell merely containing 25 mM BPO is biased from 0 to −1.7 to 0 V. However, as for the system containing the couple of 0.1 mM 4CzIPN/25 mM BPO in solution, ECL appears at −1.48 V and then reaches an extremely high peak value at ca. 1.7 V. Such phenomena well-confirm that it is reductive–oxidation ECL emission. Moreover, we notice that the peak potential of cathodic current of the 4CzIPN/BPO couple is located at −1.2 V, which is distinctly lower than that of pure 4CzIPN, i.e., ca. −1.5 V. Previously, Zu et al. also observed such a kind of feature and attributed it to the electrocatalysis effect of the coreactant on luminophores [44]. In other words, it is highly possible in that strong charge-transfer actions between those two species would be involved in such cathodic scanning. Despite this phenomenom, we notice that the resultant cathodic ECL onset potential is still as high as −1.48 V, which is followed by a quick increase in ECL intensity at the much higher potential. It thus well-confirms that both sufficient electroreduction of BPO and 4CzIPN are indispensable for such ECL emission. As calculated, such 0.1 mM 4CzIPN/25 mM BPO coreactant ECL system shows a high relative Φ_ECL_ of 197% (vs. Ru(bpy)_3_/TPrA reference, taken as 100% as the standard), which is higher than its corresponding oxidative–reduction Φ_ECL_ using the couple of 4CzIPN/TPrA, i.e., 12.0%. We speculate that the most possible reason accounting for such differences in Φ_ECL_ is the different electrochemical activity and/or reversibility of 4CzIPN^●+^ and 4CzIPN^●−^. As for the CV and ECL stability situations of such an efficient 4CzIPN/BPO couple, we performed the successive CV and ECL scanning (26 cycles). As shown in Figure 4c, the cathodic current is not stable. On increasing the scanning cycles, the cathodic current is gradually reduced, along with a monotonic shift of peak potential to the much lower value. The detailed mechanism is still unclear and in study. However, the corresponding ECL intensity seems stable enough. Especially, after the first seven cycles, the subsequent ECL intensity tends to be stable, without any noticeable fluctuations in ECL intensity.

Oxidation–reduction ECL is further constructed by using the couple of 0.1 mM BPAPTC/25 mM TPrA. As observed, it displays the typical coreactant ECL behavior. No detectable ECL can be observed while the system merely contains BPAPTC or TPrA. It is only in the presence of both BPAPTC and TPrA that we can see significant ECL emission (Figure 5a,b). Very meaningfully, the ECL signals begin to rise at an onset potential of 0.46 V and then reach the first peak value at +0.62 V and the second peak value at +1.19 V. Such ECL onset potential is distinctly lower than that of the 4CzIPN/TPrA couple shown in Figure 3 and even the lowest results among all ever reported ECLs using TADF emitters [12,18,19,29,30,31,33,34]. We speculate that the achievement of such low ECL onset potential is mainly attributed to the distinctly lowered electrochemical oxidation potentials of BPAPTC (Appendix A). Moreover, oxidation potentials of BPAPTC and TPrA are closely matched with each other, which is beneficial for coreactant ECL emission at a much lower potential. As calculated, such 0.1 mM BPAPTC/25 mM TPrA coreactant ECL system shows a high relative Φ_ECL_ of 116% (vs. 100% for the Ru(bpy)_3_/TPrA reference). Moreover, it is interesting to observe two ECL peaks, which were similar to the well-known Ru(bpy)_3_^2+^/TPrA system [44,45]. As disclosed, the evolution of those two ECL peaks were dependent on the concentrations of Ru(bpy)_3_^2+^ and TPrA. These different experimental conditions triggered different subprocesses of these coreactant systems, which well-disclosed the origins of those observed two ECL peaks [45]. The similar mechanism studies are in process for this BPAPTC/TPrA couple, which will be disclosed elsewhere. As shown in Figure 5c, the anodic CV scanning for such a couple is stable under 30 cycles, which is accompanied with stable ECL emissions shown in Figure 5d. It is believed that it is attributed to the high reversibility of BPAPTC under the electrochemical oxidation process (Appendix A).

### 3.4. ECL Spectra and Mechanisms of Coreactant TADF-ECL

Figure 6 depicts the PL and ECL spectra of these two TADF lumiphores. It is obvious that the PL and ECL spectra of the same TADF luminophores are basically identical. Compared to PL spectra in DCM, the ECL spectra of those coreactant ECL couples show slight redshifts, which is due to some extent of polarity effect of the supporting electrolyte, i.e., TBAPF_6_ (see Appendix A). To directly confirm the TADF emission nature of the coreactant ECL mode involving TPrA or BPO, the corresponding ECL spectra should be identical to the intrinsic PL spectra of those TADF luminophores concerned. In this sense, current results surely confirm this point. Previously, we noticed that the coreactant ECL spectra in TADF-polymer-modified GCE configurations [19,30] or aqueous ECL using TADF aggregates or TADF nanoencapsulation emitters [12,33,34] were significantly different from their PL counterparts. In general, a noticeable difference in λ_peak_ and/or full width at half maximum (FWHM) of the PL and ECL spectra was observed in those reports, which were attributed to some interrupting factors, such as difference in polarization [46], the involvement of surface state transition [47] or others. To rule out these interrupting possibilities, the implementation of such simplified solution-state coreactant ECL studies is very meaningful, which directly confirms the effectiveness of TADF emission in such coreactant ECL driving methods for the first time.

The mechanisms of coreactant ECL featuring TADF emission (TADF-ECL) are schematically shown in Figure 7, which is based on the above mentioned ECL studies, ECL spectra and the earlier disclosed mechanisms of TPrA- or BPO-involved coreactant ECL [19,25,48]. For the oxidative–reduction TADF-ECL, we use BPAPTC/TPrA TADF-ECL system as the example (shown in Figure 7a). Under electrochemical oxidation, holes are directly injected into the highest occupied molecular orbitals (HOMO) of BPAPTC molecules and electrons are injected from TPrA^●^ into the lowest unoccupied molecular orbitals (LUMO). After that, the columbic interactions between holes and electrons in BPAPTC lead to the generation of excitons, which is followed by ECL emission via the TADF mechanism. For the reductive–oxidation TADF-ECL, we thus use the 4CzIPN/BPO TADF-ECL system as the example (shown in Figure 7b). Under the electroreduction process, electrons are directly injected into the LUMO of 4CzIPN molecules and holes are indirectly injected from C_6_H_5_CO_2_^●^ radical intermediate species into the HOMO of 4CzIPN molecules. Subsequently, excitons (Frankel type) are generated on those 4CzIPN molecules, which is followed by ECL emission via the TADF mechanism. Thanks to the all-exciton-harvesting superiority of TADF emission, both electrochemically generated singlet and triplet excitons can radiatively decay via such coreactant TADF-ECL mode. It is thus very meaningful to further develop such coreactant TADF-ECL featuring low ECL potentials and high Φ_ECL_ towards a satisfactory application in a wide range.

## 4. Conclusions

In conclusion, we present a details study on liquid ECL using two common TADF molecules in an organic medium. Both CV, annihilation and coreactant ECL mode are measured and discussed, with the purpose of establishing more clear relationships between potential/activity/stability of those different radical intermediate species of TADF molecules and the resultant ECL potential, efficiencies, i.e., Φ_ECL_, and stabilities in coreactant ECL mode. The conclusions are as follows: (i) it is highly feasible to realize a low ECL potential for coreactant TADF-ECL by choosing TADF luminophores featuring low redox potential. In this work, a satisfactory ECL onset potential as low as +0.46 V (vs. Ag/Ag^+^) is achieved for BPAPTC; (ii) redox reversibility of TADF molecules highly determines the activity/stability of its electrogenerated radical intermediate species and the resultant ECL efficiency; (iii) if radical intermediate species of TADF luminophores are unstable, e.g., 4CzIPN^●+^, its ECL efficiency and stability can be largely promoted by using coreactant ECL mode with a suitable coreactant. Moreover, for the first time, it is confirmed that the ECL spectra of those coreactant ECL systems are identical to the PL spectra of those TADF luminophores. It thus proves that those coreactant ECL systems emit light via those TADF emitters themselves, rather than other complexed emission routes. It is anticipated that the performance and application of TADF-ECL is promising. Currently, more in-depth mechanistic studies concerning TADF-ECL have been performed and will be disclosed elsewhere.

## Data Availability

Not applicable.

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
