# Peer review of "Studies on Annihilation and Coreactant Electrochemiluminescence of Thermally Activated Delayed Fluorescent Molecules in Organic Medium"

_molecules, 2022, doi:10.3390/molecules27217457_

Round 1

Reviewer 1 Report

Thermally activated delayed fluorescent ECL (TADF-ECL)is an newly emerging concept and confirmed to be promising in the topic of ECL. Very meaningfully, it is characteristic of the capability of all exciton utilization for realizing the much higher ECL efficiency. This manuscript systematically characterized and discussed the liquid electrochemistry and ECL behavior of two typical TADF molecules in organic medium. They presented that the ECL spectra of them (coreactant ECL mode) are the same as their PL spectra counterparts, which confirms the TADF photophysical properties in the coreactant ECL mode. They also studied on the relationship between electrochemical and physical properties of TADF compounds and ECL behaviors (efficiency, potential, stability and so on). These studies are thorough and well interpreted. It can be accepted after some minor revisions.

(1). For Figure 5d, the horizontal coordinate should be changed from “Potential /vs Ag/Ag+ ” to “Time/s”.

(2) Some written format should be improved. e.g.

“very rare” to replace “extremely rare” in page 1.;

“Up to date” to replace “Up-to-date” in page 1.

“a great deal of applications has...” to replace “a great deal of applications have...”in page 1.

“such strategy” to replace “such a strategy” in page 2.

“it is believed in that” to replace “it is believed that” in page 3.

“it is believed in that” to replace “it is believed that” in page 3.

benzoyl peroixide to replace“benzoyl peroxide”in page 4.

that such unstable”to replace“that such an unstable”in page 7.

such kind of to replace“such a kind of ”in page 8.

are basically the identical...”delete “the”in page 9.

in wide range to replace“in a wide range”in page 10.

Author Response

(1). For Figure 5d, the horizontal coordinate should be changed from “Potential /vs Ag/Ag+ ” to “Time/s”.

Response: Thanks for your suggestion. In the revised formation, we have corrected it as you mentioned (highlighted as red colors).

(2) Some written format should be improved. 

Response: Thanks for your so kindly suggestion, we have improved those written issues.

Reviewer 2 Report

This work is original and relevant to the topic of ECL. It is very meaningful to explore qualified organic ECL luminophores to develop the ECL technique. This manuscript gave a detailed and in-depth discussion about thermally activated delayed fluorescent ECL (TADF-ECL) in the organic solvent. It seems interesting and important since the topic of TADF-ECL is a novel concept that was confirmed to be a very promising strategy to enhance the ECL efficiency of organic ECL luminophores. In this emerging new topic, these authors are one of the main contributors. These authors presented the detailed electrochemistry and ECL studies (annihilation ECL and coreactant ECL) of two TADF compounds in the organic medium and confirmed that the ECL spectra of coreactant ECL mode can be identical to its PL spectra for the first time. The corresponding experiments and discussion are impressive and reasonable. Accordingly, I think it is a good original research paper. After a minor revision, it can be accepted in this journal.

Minor Revision suggestions:

1. The language needs some minor improvement. For example, some sentences or words should be improved further, i.e. A sentence in lines 15-16. “overlap with” (line 18), experimentally (Line 20), “relied on” (line 32)

2. For Scheme 1, it is suggested to give more explanation about those abbreviations, such as ISC, RISC, PF-ECL, and RISC. Such physical processes are not easy to understand for the general reader in ECL.

Author Response

Thanks very much for your minor revision suggestion. The point-by-point response is listed below:

1)1. The language needs some minor improvement. For example, some sentences or words should be improved further, i.e. A sentence in lines 15-16. “overlap with” (line 18), experimentally (Line 20), “relied on” (line 32)

Response: Thanks for your such advisement. In the revised format, we have corrected these mentioned language issues (which is highlighted as red colors in the revised manuscript).

2. For Scheme 1, it is suggested to give more explanation about those abbreviations, such as ISC, RISC, PF-ECL, and RISC. Such physical processes are not easy to understand for the general reader in ECL.

Response: In the revised version, we have added more explanations for those abbreviations.

Reviewer 3 Report

This manuscript gave an impressive illustration about thermally activated delayed fluorescence (TADF) electrochemiluminescence (TADF-ECL) either in annihilation or coreactant mode. This is a significant study since the exploration of such novel kind of pure organic ECL luminophores is important towards solving the challenges in ECL efficiency. As one main researcher in TADF-ECL, in this manuscript, this group discussed the relationship between between potential/activity/stability of different radical intermediate species of TADF molecules and the resultant ECL potential, efficiencies, and stabilities in coreactant ECL mode. Moreover, they confirmed that ECL spectra in coreactant ECL mode and its corresponding PL spectra are identical, which supported the validity of TADF-ECL in coreactant mode. All experiments and analysis are in-depth and self-consistent. Therefore, it is a well-written paper and I recommend it can be accepted for publication. Some minor revisions are suggested as follows:   

1) There are some editing errors that need to be corrected and checked. For example, the words “oxidation-reduction ECL” shown in Line 321. It is suggested to be “Oxidative-reduction ECL”; 0.1 M 4CzIPN (Line 262) and 0.1 M 4CzIPN/40 mM TPrA(Line 256) should be further checked since the concentration of 4CzIPN shown in caption of Figure 3 is 0.1 mM.

2) Following the question 1, I suggest the author further optimize the language and overcome other unmentioned typographical errors.

Author Response

Thanks for your so kindly suggestions for our manuscript. The point-by-point revisions according to your questions are listed below:

1) There are some editing errors that need to be corrected and checked. For example, the words “oxidation-reduction ECL” shown in Line 321. It is suggested to be “Oxidative-reduction ECL”; 0.1 M 4CzIPN (Line 262) and 0.1 M 4CzIPN/40 mM TPrA(Line 256) should be further checked since the concentration of 4CzIPN shown in caption of Figure 3 is 0.1 mM.

Response: In the revised part, we have corrected these editing error. All those revisions are highlighted as red color).

2) Following the question 1, I suggest the author further optimize the language and overcome other unmentioned typographical errors.

Response: For the revised manuscript, we have further optimized the language and removed other unmentioned editing errors. (highlighted as red colors).